# Transcriptional bursting is intrinsically caused by interplay between RNA polymerases on DNA

Keisuke Fujita[1,2], Mitsuhiro Iwaki[1,2] & Toshio Yanagida[1,2]

Cell-to-cell variability plays a critical role in cellular responses and decision-making in a population, and transcriptional bursting has been broadly studied by experimental and theoretical approaches as the potential source of cell-to-cell variability. Although molecular mechanisms of transcriptional bursting have been proposed, there is little consensus. An unsolved key question is whether transcriptional bursting is intertwined with many transcriptional regulatory factors or is an intrinsic characteristic of RNA polymerase on DNA. Here we design an *in vitro* single-molecule measurement system to analyse the kinetics of transcriptional bursting. The results indicate that transcriptional bursting is caused by interplay between RNA polymerases on DNA. The kinetics of *in vitro* transcriptional bursting is quantitatively consistent with the gene-nonspecific kinetics previously observed in noisy gene expression *in vivo*. Our kinetic analysis based on a cellular automaton model confirms that arrest and rescue by trailing RNA polymerase intrinsically causes transcriptional bursting.

[1] Laboratory for Cell Dynamics Observation, Quantitative Biology Center, RIKEN, 6-2-3, Furuedai, Suita, Osaka 565-0874, Japan. [2] Soft Biosystem Group, Laboratories for Nanobiology, Graduate School of Frontier Biosciences, Osaka University, 1-3 Yamadaoka, Suita, Osaka 565-0871, Japan. Correspondence and requests for materials should be addressed to K.F. (email: keisuke.fujita@riken.jp) or to T.Y. (email: yanagida@fbs.osaka-u.ac.jp).

Genetically identical cells in an identical environment behave differently, leading to significant consequences in many biological process from bacterial decision-making to mammalian development[1–4]. The well-known potential source of this cell-to-cell variability is a noisy messenger RNA production in transcription, which is the so-called 'transcriptional bursting', and this mechanism has been broadly studied by theoretical and experimental approaches. According to the theoretical studies, transcriptional bursting can be explained by a two-state model of gene regulation, where a gene switches between on and off states at constant rates ($k_{on}$ and $k_{off}$) and mRNA is produced at a constant rate ($k_T$) in the on-state[5]. In experimental studies, single-molecule fluorescence in situ hybridization (FISH) data demonstrated the gene-nonspecific trend between mean and a variety of mRNA distributions in individual cells[6,7]. This gene-nonspecific trend in a bacterial cell has been observed in many eukaryotic cells including mammalian cells[8]. However, there is no clear molecular model to explain the transcriptional bursting in spite of advanced understanding of the kinetics in a cell. Recent studies of the molecular mechanisms have shown that supercoiling accumulation on a constrained DNA[9] and modulation of the promoter structure[10] affect the kinetics of transcriptional bursting. These different models suggest a complicated mechanism intertwined with many transcriptional regulatory factors in a cell.

Here we design a bottom-up approach to examine whether transcriptional bursting requires such a complicated molecular mechanism. Our results show transcriptional bursting occurs with the minimum components of bacterial transcription, suggesting an intrinsic molecular mechanism of transcriptional bursting. Based on the kinetic analysis, we propose that transcriptional bursting is intrinsically caused by interplay between RNA polymerases (RNAPs) on DNA. Our analysis can quantitatively explain the gene-nonspecific kinetics previously observed in noisy gene expression in vivo[6].

## Results

**Visualization of the on–off switching in mRNA production.** Our experimental system is based on a fastFISH method[11], which is capable of visualizing the on–off switching of transcription in real time (Fig. 1a). In the experimental setup, a bacterial RNAP binds to the T7A1 promoter on a 546 bp DNA template (Fig. 1b), which is tethered to a poly(ethylene glycol) (PEG)-coated glass surface via biotin–avidin interaction and initiates transcription elongation under the microscope. A nascent mRNA molecule was visualized by a self-quenched oligo probe modified with Cy3 and quencher, with the hybridization rate of $0.2\,s^{-1}$ at our experimental conditions (Supplementary Fig. 1). The position of the DNA template was detected by Cy5 spot and mapped onto

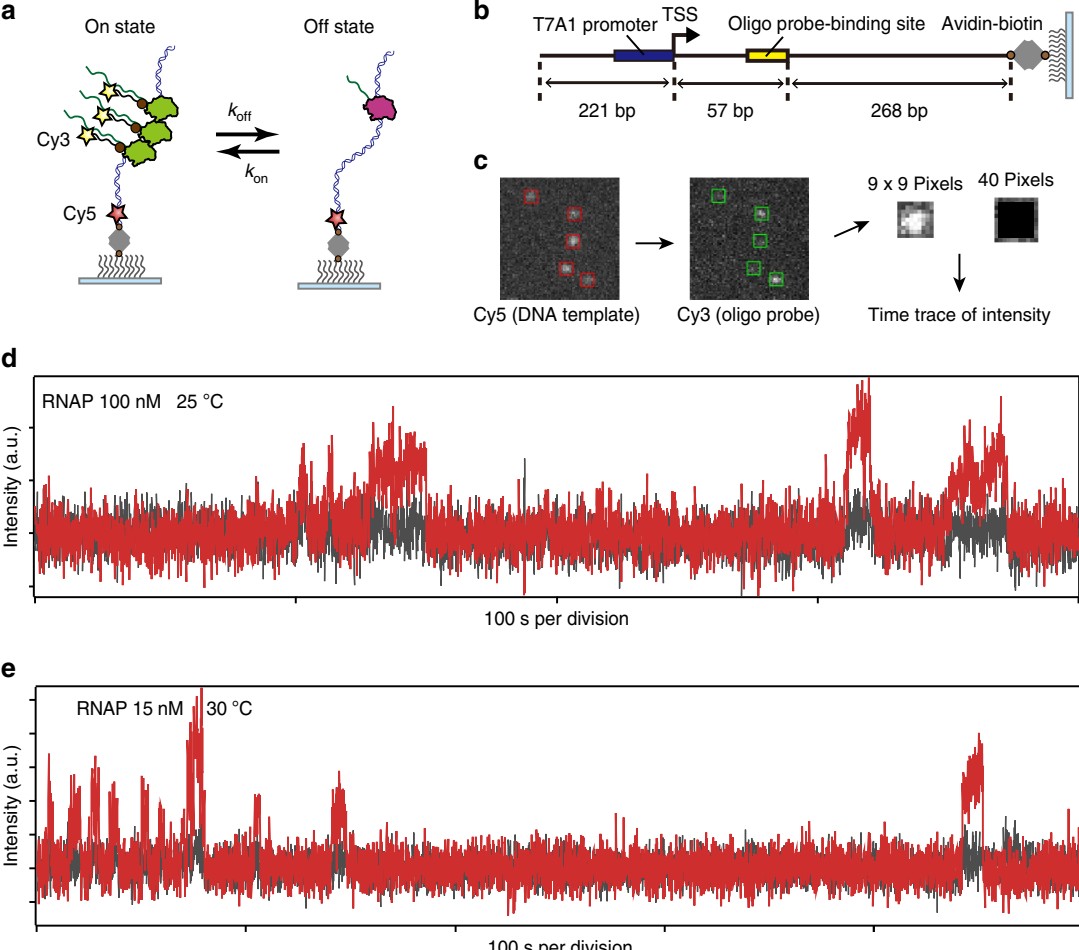

**Figure 1 | Reconstruction and visualization of transcriptional bursting. (a)** Experimental design. The on–off switching in transcription is monitored by a Cy3-modified oligo probe in real time. **(b)** Schematic of DNA template. **(c)** Analysis procedure (Methods). **(d,e)** Representative traces of Cy3 fluorescence intensity. The red and black lines indicate the intensity of ROI and its perimeter, respectively. Consecutive oligo probe bindings were interrupted by off states, corresponding to burst-like mRNA production. The experiments were performed at different RNAP concentrations and temperatures.

Cy3 channel (Fig. 1c). By calculating the background-corrected fluorescence intensity of Cy3 at the position of the DNA template in each recording frame (Methods), we obtained time traces of mRNA production at different RNAP concentrations and temperatures. The experiments were performed at the nucleotide concentration of 100[NTP] (1 mM GTP and UTP, 500 μM ATP and 250 μM CTP), to correct for the biased transcription rate for each nucleotide species[12]. Surprisingly, the time traces show burst-like mRNA production, suggesting off-states during transcription even in the simplest *in vitro* transcription system (Fig. 1d,e).

**The off-state by arrest during transcription elongation**. This *in vitro* experimental system visualizes only mRNA production and it is difficult to clarify what mechanism causes the off-state. To access the molecular dynamics of RNAP simultaneously with mRNA production, we designed another experimental setup, which is capable of visualizing the translocation of RNAP and mRNA production simultaneously. In this experiment, we prepared an open complex by incubation of a quantum dot (Qdot)-labelled RNAP holoenzyme and a DNA template containing a *lac*UV5 promoter and single- or six-target sites for the oligo probe (Fig. 2a). The DNA–RNAP complex was attached to the glass surface by chamber flow via biotinylation at both ends of the DNA template. A Qdot-labelled RNAP then starts elongation by addition of nucleotides along a stretched DNA template under the microscope (Fig. 2b). Tracking of Qdot-RNAP and detection of mRNA production were performed simultaneously (Fig. 2c). Tracking trace of a Qdot-RNAP shows a linear movement is followed by oligo-binding events and a linear movement indicates elongation. The nucleotide concentration dependency of velocity was consistent with the previous report[12] (Supplementary Fig. 2a–c), suggesting the setup of the experiment does not

impede the elongation process. The traces in our experiment always showed an irreversible long pause of RNAP after linear movement in a recording time (20 min). The displacement between the first position of RNAP and the last position rarely reached the full-length of mRNA (100 nm), as shown in the histogram of displacement at 100[NTP] (Fig. 2d). In addition, this broad displacement was observed on a DNA template containing a T7A1 promoter (Supplementary Fig. 2d). We considered the irreversible long pause as arrest, which is reported by an *in vitro* single-molecule measurement (Supplementary Fig. 3a)[13,14]. Assuming that RNAP is arrested with a constant rate at each base pair, we estimated that the arrest rate ($k_A$) is $\sim 0.2\,s^{-1}$ (Supplementary Fig. 3b). Finally, we concluded that the off-state in our *in vitro* single-molecule measurements originated from an arrest state. This was supported by atomic force microscopy (AFM) images of transcription elongation as well (Fig. 3).

**Transcription initiation dependence of bursting**. According the previous biochemical assay[15], an arrested RNAP is rescued by a trailing RNAP; therefore, this arrest and rescue can be the on–off switching in transcription bursting. To extract the kinetics of the on–off switching from the time trace of mRNA production (Fig. 1d,e), we counted the number of oligo probe-binding events on a single DNA molecule in a 400 s recording time at different RNAP concentrations and temperatures with the T7A1 promoter (the template is shown in Fig. 1b). The normalized histogram of detected mRNA molecules showed dependence on RNAP concentrations and temperatures, and therefore the rate of open complex formation, which is uniquely determined by these two parameters (Supplementary Fig. 4). This dependency suggests that the open complex formation is a rate-limiting step of the transcription initiation in our experimental design. Here we defined the open complex formation rate as the transcription

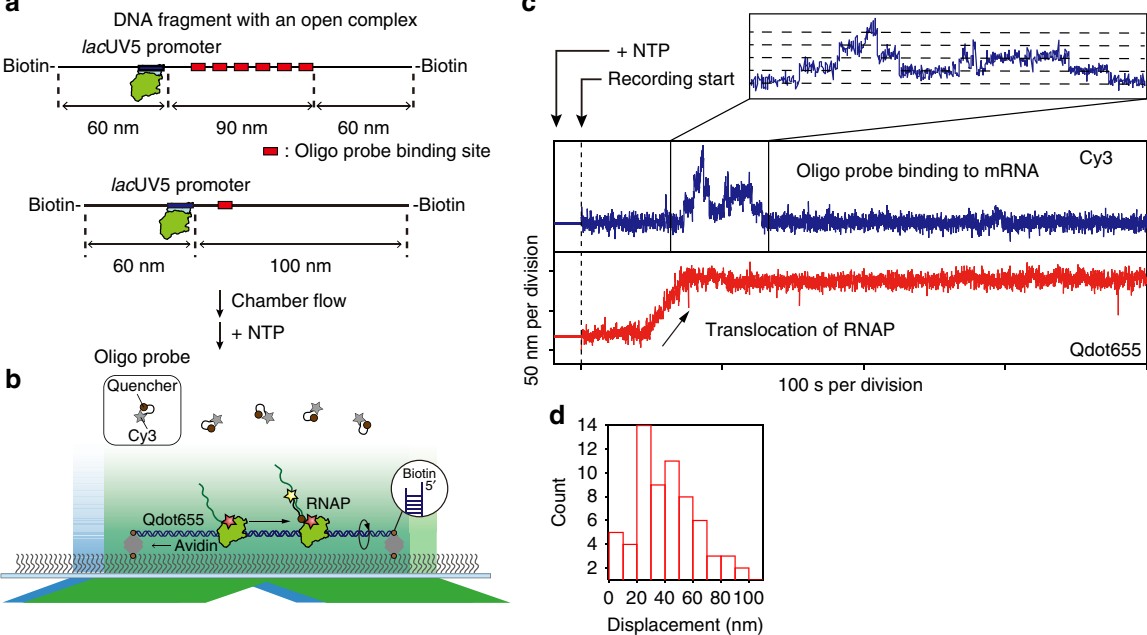

**Figure 2 | Arrest on an unconstrained DNA. (a)** Schematic of DNA templates (six target and single target). **(b)** A Qdot-labelled RNAP starts elongation by addition of nucleotides along a stretched DNA template under TIRF microscope. A nascent mRNA molecule was visualized by a self-quenched oligo probe-modified Cy3. As a single biotin molecule anchors the DNA at each end, the DNA template can rotate around its linkage. **(c)** Simultaneous observation of RNAP translocation and mRNA production in real time with the six-target template. RNAP translocation was followed by stepwise increases of fluorescence intensity of Cy3, corresponding to transcription elongation. **(d)** Distribution of displacement during elongation at 100[NTP] with the single-target template. It is noteworthy that our analysis possibly underestimates the number at the first few bins (see Methods). $N = 67$. Data were obtained from four independent successfully reproduced experiments.

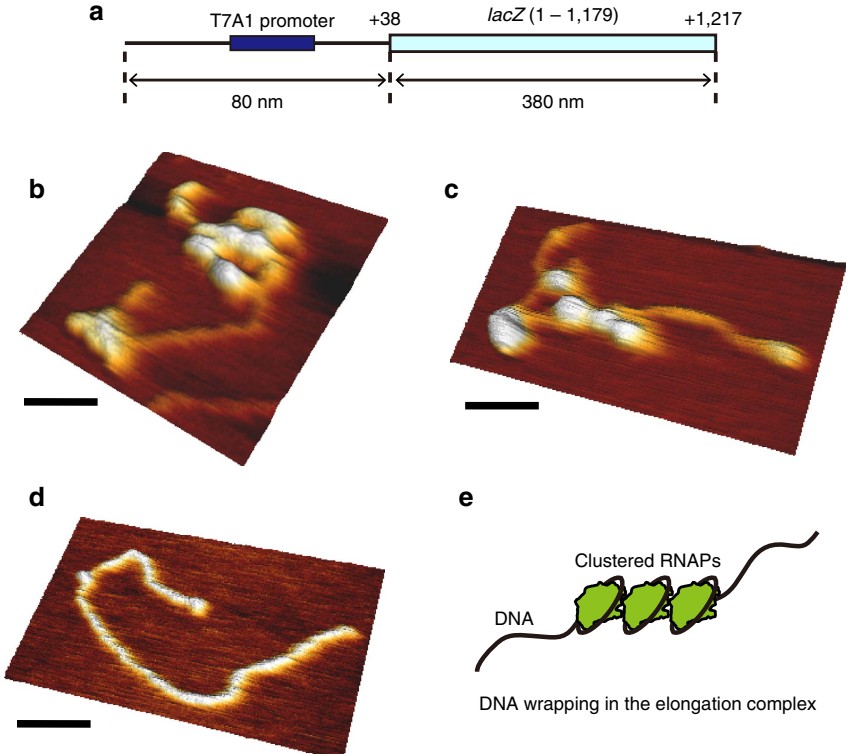

**Figure 3 | AFM image of clustered RNAPs by arrest.** (**a**) Schematic of 1.4 kb DNA template. The DNA template contains a T7A1 promoter and *lacZ* (1–1179). +38 and +1217 indicate the relative positions from the transcription start site. (**b,c**) Representative AFM images of a 1.4 kb DNA template with RNAP during transcription. RNAP molecules locally accumulate on the DNA and form a cluster. AFM images were acquired on mica in air. The ratio of a DNA template with RNAP was $0.75 \pm 0.03$ ($\pm$s.e.m., $N = 184$), giving $k_A = 0.01\,\mathrm{s}^{-1}$ (Supplementary Fig. 3b). (**d**) An AFM image of a bare DNA template as a negative control. (**e**) DNA wrapping in the elongation complex[35] explains the shorter contour length of DNA in **b,c** than in **d**. Scale bars, 50 nm. Data were obtained from three independent successfully reproduced experiments.

initiation rate. Figure 4a–j shows the histograms of detected mRNA molecules at the different transcription initiation rates. At low initiation rates the histograms can be described by a Poisson distribution, whereas at high initiation rates the histograms deviate from a Poisson distribution. This deviation from a Poisson distribution is characterized by a zero peak and this so-called 'Poisson with zero spike' distribution of mRNA copy number is observed *in vivo* as well[9]. We evaluated the initiation rate dependency of the mean and the Fano factor (the ratio between the variance and the mean) for the distributions. The Fano factor indicates the deviation from a Poisson distribution and it equals 1 for a Poisson distribution. As the plots show (Fig. 4k,l), the mean increases with transcription initiation rate and the Fano factor increases with mean. An increase of the mean with transcription initiation rate can be described by a Hill equation with coefficient $1.9 \pm 0.2$ ($\pm$s.d.), suggesting cooperativity in transcription. Furthermore, in the relationship between the Fano factor and mean, we found that the Fano factor changes as the mean to the power $0.71 \pm 0.25$ ($\pm$s.d.) (Fig. 4l and Supplementary Fig. 5), which is consistent with the mean to the power 0.64 observed on single-molecule FISH data *in vivo*[6]. This consistency supports the interpretation that the arrest and rescue correspond to the on–off switching in transcriptional bursting and underlie the gene-nonspecific constraint in transcriptional bursting of bacteria, and possibly eukaryotic cells[8].

**Analysis of bursting by the cellular automaton model.** To describe the trend between the mean and the Fano factor by the molecular dynamics of RNAP (arrest and rescue), we performed a simulation based on a cellular automaton model (Fig. 5 and Supplementary Software 1)[16,17]. In the model, a DNA template was represented by 325 boxes aligned in one dimension from 3' (left)-end to 5' (right)-end, with the end of the oligo-binding site at box 57 (Fig. 1b) and where RNAP moves according to the following rules: RNAP moves unidirectionally from left to right. When the space between two adjacent RNAPs is <15 boxes, the situation is regarded as a collision. The 15 boxes were estimated by analogy of eukaryotic RNAP collision[18], considering the smaller interaction between bacterial RNAP and DNA during elongation. If no collision occurs, RNAP moves to the next box at a rate of $k_F/(k_F + k_A)$ or RNAP enters an arrest state at a rate of $k_A/(k_F + k_A)$. In this model, one simulation time is consistent with $1/k_F = 100\,\mathrm{ms}$. Arrest state is rescued by pushing of a trailing RNAP (collision) at a rate of $k_R$. When RNAP moves from the 71th and 72th box, the 57th box of the nascent mRNA is 'exposed' and the mRNA is 'visualized' and counted (here we assumed that 14 boxes of the nascent mRNA are protected by RNAP[19]). RNAP appears in the first box at a rate of $k_I$. The $k_A$ and $k_R$ in the model were decided by comparison with the mean and Fano factor of the experimental distribution at $k_I = 0.045\,\mathrm{s}^{-1}$ (Fig. 4f and Supplementary Fig. 6). The trend was reproduced when $k_A$ and $k_R$ were 0.05 and 0.003 s$^{-1}$, respectively (Fig. 4l and Supplementary Figs 6 and 7). On the other hand, in the plot of mean and initiation rate, the simulated data was described with a Hill coefficient, $1.0 \pm 0.2$ ($\pm$s.d.) and was not able to reproduce the cooperativity (Fig. 4k). Furthermore, we estimated that $k_A$ is $0.00005\,\mathrm{s}^{-1}$ more than enough to reproduce transcriptional bursting *in vivo* based on the cellular automaton model (Fig. 6 and Supplementary software 2) as $k_R$ is fixed at $0.003\,\mathrm{s}^{-1}$, corresponding to $k_{on} = 0.003\,\mathrm{s}^{-1}$ and $k_{off} = 0.0004\,\mathrm{s}^{-1}$ in the two-state model. These parameters are comparable to the reported value *in vivo*[6]. Thus, our simple

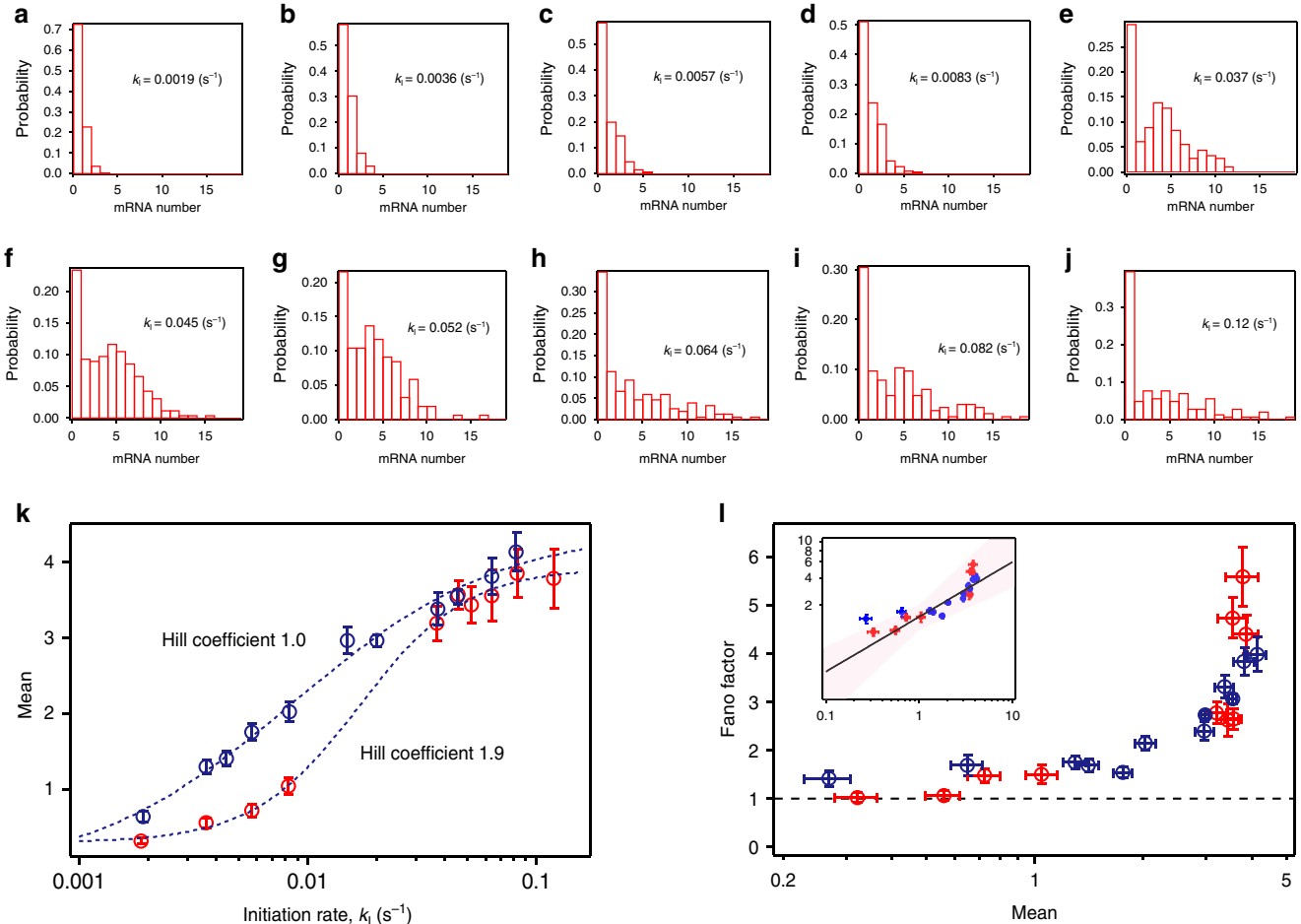

**Figure 4 | Cooperative and non-Poisson mRNA production.** (**a**–**j**) mRNA number distributions obtained by counting the number of oligo probe binding events in a given time (400 s). The distributions depend on transcription initiation rate $k_I$. $N = 184$, 161, 190, 206, 179, 256, 153, 150, 164 and 141 in **a**–**j**. (**k**) Mean of mRNA number distribution versus $k_I$. The plots can be described by a Hill equation (dashed line) with Hill coefficient, $1.9 \pm 0.2$ ( $\pm$ s.d.). The blue plot indicates the simulated data, which can be described with Hill coefficient, $1.0 \pm 0.2$ ( $\pm$ s.d.). (**l**) Fano factor versus mean of mRNA number distribution. The dashed line indicates pure Poisson distribution. The inset shows the log–log plot (see Supplementary Fig. 5 for the details). Error bars indicate the bootstrapped s.d. in **k**,**l**. All data sets were obtained from more than three independent successfully reproduced experiments.

model explains the kinetics of transcriptional bursting by linking the arrest and rescue to the on–off switching in the two-state model (Fig. 7a,b).

## Discussion

To explain the discrepancy of the Hill coefficients between experimental data (1.9) and our model (1.0), however, we may need other sources of cooperativity such as indirect interplay via DNA stress generated by RNAP. As previously reported, RNAP generates local torsional stress on DNA, which is sufficiently long-lived to untwist the upstream region even on linear DNA[20] and supercoiling affects the transcription initiation rate[21]. Furthermore, this indirect interplay between RNAP molecules through supercoiling may explain why the arrest rate ($k_A$) varies by a few orders of magnitude depending on measurement conditions not only in this study ($k_A = 0.00005 \sim 0.2\,\text{s}^{-1}$), but also in the previous study ($k_A = 0.0012 \sim 0.02\,\text{s}^{-1}$; Supplementary Fig. 3 and Supplementary Table 1), as the kinetics of transcription elongation is modulated by torque via DNA[22]. The interpretation that $k_A$, that is $k_{off}$ (Fig. 7b), was largely modulated by torsional stress on DNA is compatible with the previous model that argues that DNA supercoiling affects the kinetics of transcriptional bursting[9]. Furthermore, the largely changeable $k_A$, relative to the other parameters ($k_R$ and $k_I$), is consistent with

the suggested kinetic scheme under the gene-nonspecific constraint, in which only $k_{off}$ ($10^{-5} \sim 10^2\,\text{s}^{-1}$) is varied to change the gene expression level[6]. In addition, our model is compatible with the model that promoter structure and state control the kinetics of transcriptional bursting[10,23] by the modulation of transcription initiation rate ($k_I$). The $k_I$ is a critical parameter to control not only the mean of mRNA distribution but also the Fano factor as our experimental data suggest.

At the same time, we note that models where the binding and unbinding of regulatory factors such as gyrase, and possibly enhancers in eukaryotic cells, cannot by themselves fully explain the gene-nonspecific kinetics of transcriptional bursting observed *in vivo*. In such models, where the regulatory proteins turn on the gene, the $k_{off}$ corresponds to their unbinding rate, which is generally constant as long as the binding sequence is unchanged, whereas in fact $k_{off}$ is largely changed in a gene-nonspecific manner[6]. In other words, $k_{off}$ in the previous models must be gene specific, which is contrary to the global feature of transcriptional bursting observed *in vivo*. On the other hand, to explain the gene-specific feature of transcriptional bursting as observed in yeast[8], such regulatory proteins may be required.

In the cellular automaton model, we completely excluded the effect of DNA sequence and regulatory proteins to reproduce our

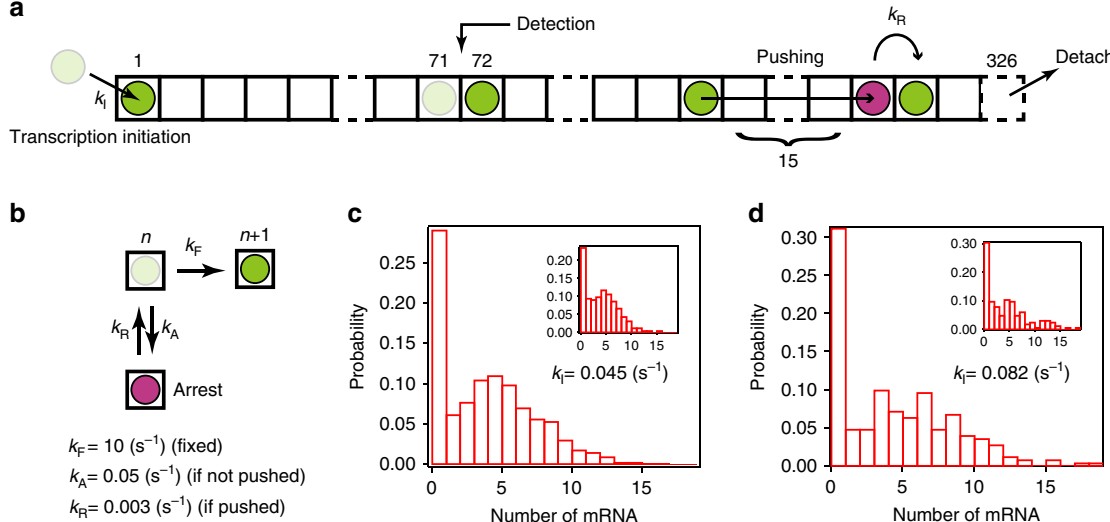

**Figure 5 | A stochastic cellular automaton model of transcription.** (**a,b**) Schematic of a stochastic cellular automaton model. A DNA template was represented by 325 boxes aligned in one dimension from 3′ (left) end to 5′ (right) end. RNAP represented by a ball, moves according to the rule as follows: RNAP moves unidirectionally from left to right. When the space between two adjacent RNAPs is <15 boxes, the situation is regarded as a collision. If no collision occurs, RNAP moves to the next box at a rate of $k_F/(k_F + k_A)$ or RNAP enters an arrest state (red ball) at a rate of $k_A/(k_F + k_A)$. In this model, one simulation time is consistent with $1/k_F = 100$ ms. Arrest state is rescued by pushing of a trailing RNAP (collision) at a rate of $k_R$. According to the actual DNA template (Fig. 1b), when RNAP moves from the 71th and 72th box, the nascent mRNA is 'visualized' and counted. RNAP appears in the first box at a rate of $k_I$. (**c,d**) mRNA number distributions obtained by the number of counted mRNAs in 400 s. $N = 1,250$ for **c** and $N = 250$ for **d**. For comparison, the inset shows the experimental data corresponding to the same value of $k_I$.

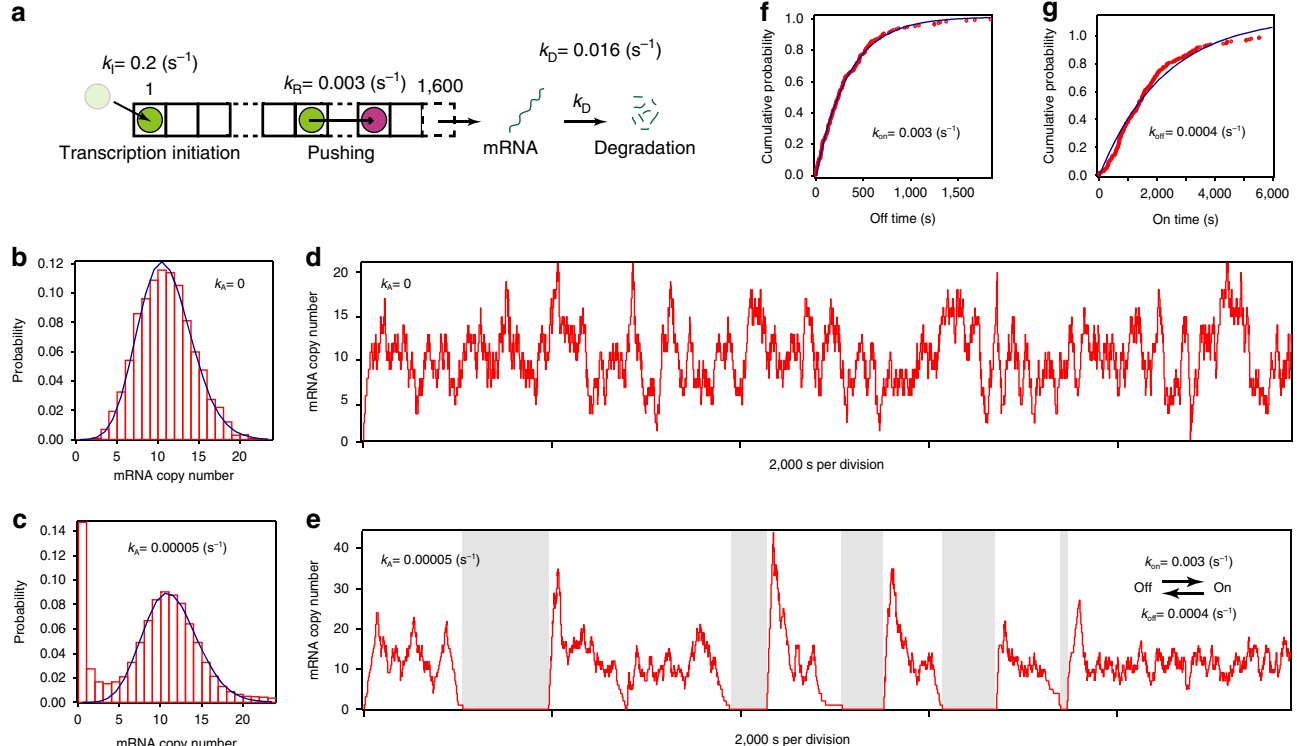

**Figure 6 | Transcriptional bursting caused by arrest and its rescue.** (**a**) A stochastic cellular automaton model with mRNA degradation. The length is 1,600 boxes, which is the average length of an operon in *E. coli*[36]. When RNAP moves to the last box, mRNA is produced. mRNA is degraded at the rate of $k_D$ (0.016 s$^{-1}$ corresponding to a lifetime of 1 min). Except for the above, the rules are the same as Fig. 5. (**b,c**) Simulated mRNA copy number probability distribution. Distributions can be described by a Poisson distribution (blue line) in **b** and without the first five bins in **c**. The bins above 25 (4%) are not shown for simplicity in **c**. $N = 98,398$ in **b** and $N = 1,998,379$ in **c**, corresponding to the simulation time. (**d,e**) Simulated time trace of mRNA copy number. Even at a low arrest rate ($k_A \sim 0.00005$ s$^{-1}$), the trace shows a stochastic switching between on state (the number of mRNA >0) and off state (the number of mRNA = 0). (**f,g**) Cumulative frequency of dwell time for the on and off state in **c**. Dwell times were fit to a single exponential function. The calculated parameters, $k_{on}$ and $k_{off}$ are $0.0029 \pm 0.00002$ s$^{-1}$ and $0.00044 \pm 0.00001$ s$^{-1}$ ($\pm$ s.d.), respectively.

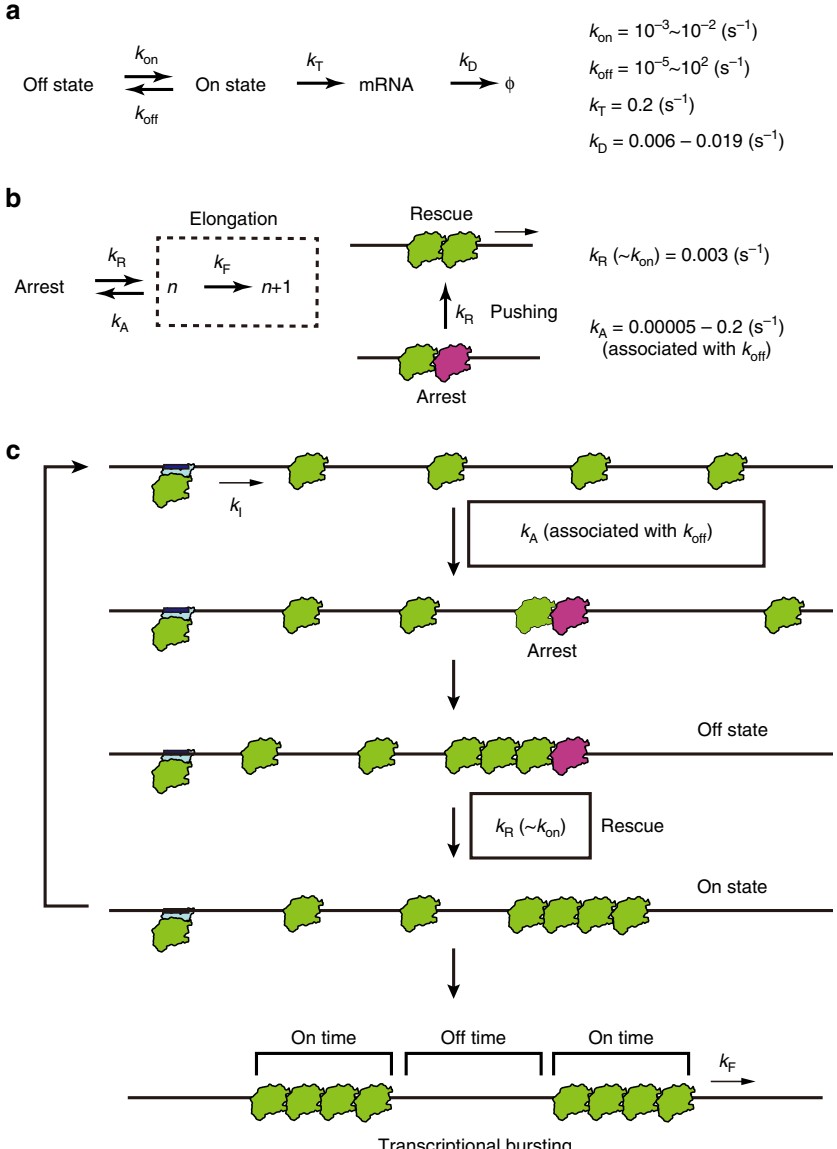

**Figure 7 | Arrest-pushing model for transcriptional bursting. (a)** Two-state model of transcription. The relevant kinetic parameters were calculated from the data of an single-molecule FISH (smFISH) experiment in bacteria[6]. **(b)** Two-state model based on single-molecule kinetics of RNAP. Arrested RNAP is rescued by pushing from a trailing RNAP corresponding to on–off switching in the two state model. **(c)** The gene-nonspecific constraint in transcriptional bursting of bacteria. DNA supercoiling and promoter structure modulate the kinetics of transcriptional bursting via transcription arrest rate ($k_A$) and initiation rate ($k_I$), respectively.

*in vitro* data, especially the trend between Fano factor and mean. However, we cannot fully rule out such effects by our *in vitro* experiments alone. Specifically, the sequence dependence of arrest rate during elongation was not fully examined, because our *in vitro* experiment (Fig. 2) does not have sufficient spatial resolution to discuss the arrest position in the context of sequence. However, the quantitative consistency between our simulated and experimental data, and between our *in vitro* data and the previous *in vivo* data (Fig. 4l and Supplementary Fig. 5) suggests that the arrest–rescue process intrinsic to RNAP and DNA is the dominant source of transcriptional bursting in bacteria.

Our model argues that, although DNA supercoiling and promoter state modulate the kinetics of transcriptional bursting through $k_A$ and $k_I$, interplay between RNAP molecules under-lies the gene-nonspecific constraint (Fig. 7c). This mechanism may seem too simple to explain the molecular mechanism of transcriptional bursting and its gene-nonspecific feature in eukaryotic cells in addition to bacteria. Although transcription is a complicated system intertwined with many regulatory factors, especially in eukaryotic cells, we note that the multi-subunit RNAPs such as RNAP, which play the central role in transcription of all organisms, are highly conserved and, presumably, the regulation of the elongation process was achieved earlier in evolution than the initiation process[24]. Furthermore, a recent study suggested that gene regulation evolved, because it increases gene expression noise and benefits an organism as a result[25]. We believe that because transcriptional bursting (gene expression noise), which leads to cell-to-cell variability, benefits an organism without regulation of initiation, it is plausible that our model is the most evolutionarily primitive mechanism associated with elongation that is responsible for the gene-nonspecific kinetics seen in contemporary organisms.

## Methods

**DNA constructs.** In kinetic measurements by fastFISH (Figs 1 and 4), we used the following sequence as a 546 bp DNA template containing a single-target site of the oligo probe.

5′-CCACAACGGTTTCCCTCTAGAAATAATTTTGTTTAACTTTAAGA AGGAGATATACATAaaagacgccttgttgttagccataaagtgataacctttaatcattgtctttatta atacaactcactataaggagagacaacttaaagagacttaaaagattaatttaaaatttatcaaaaagagtattgact taaagtctaacctataggatacttacagccatcgagagggacacggcgaatagccatcccaatcgacaCCCT ATCCCTTATCTTAACCACTCCAATTACATACACCTTTCAAAACTTCAA ACTTCAAACTTCAAACTTCAAACTTCAAACTTCAAACTTCAAACTTCA AACTTCAAACTTCAAACCACCGTTGATATATCCCAATGGCTGCATT TCAAAACTTCAAACTTCAAACTTCAAACTTCAAACTTCAAACTTCAAA CTTCAAACTTCAAACTTCAAACTTCAAACTGCAGCTGGATATTACGGCC TTTTTAAAGACCGTAAAGAAAAATAAGCACAAGTTTTATCCGGC-3′-(Bio).

In the above sequence, lower case letters indicate the T7A1 promoter region ($-163$ to $+38$ from the transcription start site represented by the bold typed a). An underline indicates the oligo probe-binding site. The template sequence from the oligo probe-binding site to the 3′-end (downstream region) is the same as in the original report[11]. The self-quenched oligo probe is 5′-/Cy3/ GTTAAGATAAGGGATAGGG/RQ/-3′ (synthesized by Integrated DNA Technologies, Coralville, IA). RQ indicates Iowa Black RQ. To construct s template DNA fragment, we inserted the extended promoter region and the downstream region between the NdeI and HindIII sites in pT7-7. The DNA templates were prepared by PCR from the plasmid with

5′-CCACAACGGTTTCCCTCTAG-3′ and

5′-/5Bio/G/Cy5/CCGGATAAAACTTGTGC-3′ (synthesized by Integrated DNA Technologies).

See Supplementary Fig. 8a,b for the structures of the modifications.

In the Qdot tracking experiment with fastFISH (Fig. 2), we used the two kinds of DNA template containing six-target sites and single-target of the oligo probe.

Six-target template (641 bp): 5′-CCACAACGGTTTCCCTCTAGAAATAAT TTTGTTTAACTTTAAGAAGGAGATATACATAtaatgcagctggcacgacaggtttc tatgcttccggctcgtataatgtgtgg**a**attgtgagcggataacaatttcacacaggaaacagctatggaccgcaagctt CCCTATCCCTTATCTTAACCACTCCAATTACATACACCCCTATCCCTT ATCTTAACCACTCCAATTACATACACCCCCTATCCCTTATCTTAACC ACTCCAATTACATACACCCCCTATCCCTTATCTTAACCACTCCAAT TACATACACCCCCTATCCCTTATCTTAACCACTCCAATTACATACA CCAGCTCCCTATCCCTTATCTTAACCACTCCAATTACATACACCTTTCA AAACTTCAAACTTCAAACTTCAAACTTCAAACTTCAAACTTCAAACT TCAAACTTCAAACTTCAAACTTCAAACCACCGTTGATATATCCCAATG GCTGCAGCTG GATATTACGGCCTTTTTAAAGACCGTAAAGAAAAATA AGCACAAGTTTTATCCGGC-3′.

In the above sequence, lower case letters indicate the lacUV5 promoter region ($-131$ to $+53$ from the transcription start site represented by the bold typed a). Six underlines indicate the oligo probe-binding sites.

Single-target template (517 bp): 5′-CCACAACGGTTTCCCTCTAGAAATAATTTTGTTTAACTTTAA GAAGGAGATATACATAtaatgcagctggcacgacaggtttcccgactggaaagcgggcagtga gcgcaacgcaattaatgtgtgagttagctcactcattaggcacccagctttacactttatgcttccggctcgtata atgtgtgg**a**attgtgagcggataacaatttcacacaggaaacagctatgCCCTATCCCTTATCTTAAC CACTCCAATTACATACACCTTTCAAAACTTCAAACTTCAAACTTCAAA CTTCAAACTTCAAACTTCAAACTTCAAACTTCAAACTTCAAACTT CAAACCACCGTTGATATATCCCAATGGCTGCATTTCAAAACTTCAAACT TCAAACTTCAAACTTCAAACTTCAAACTTCAAACTTCAAACTTC AAACTTCAAACTTCAAACTGCAGCTGGATATTACGGCCTTTTTAAA GACCGTAAAGAAAAATAAGCACAAGTTTTATCCGGC-3′.

In the above sequence, lower case letters indicate the lacUV5 promoter region ($-131$ to $+41$ from the transcription start site represented by the bold typed a). Single underline indicates the oligo probe-binding sites. We prepared the templates for biotinylation at the both ends by PCR with

5′-/5Bio/CCACAACGGTTTCCCTCTAG-3′ and

5′-/5Bio/GCCGGATAAAACTTGTGC-3′.

See Supplementary Fig. 8c for the structure of the biotin modification.

In AFM imaging (Fig. 3), we used a 1.4 kb DNA template containing a T7A1 promoter and lacZ (1-1179 when 1 indicates the first nucleotide of the gene) (Fig. 3a). The upstream region from lacZ is the same as the DNA template in fastFISH experiment. We prepared the templates by PCR with

5′-CCACAACGGTTTCCCTCTAG-3′ and

5′-ATAATGCGAACAGCGCAC-3′.

PCR-amplified DNA fragments were purified by Wizard PCR Preps DNA Purification System (Promega). The sequence of the DNA template region in plasmids was verified by sequencing. Except as otherwise noted, the primers were synthesized by Hokkaido System Science Co., Ltd, Japan.

**Protein expression and purification.** For the fastFISH and AFM imaging experiment, we used Escherichia coli RNAP holoenzyme purchased from New England Biolabs. For the Qdot-tracking experiment, we designed the plasmid for co-overexpression of tagged E. coli RNAP subunits[26]. To construct the plasmid, we inserted the tandem genes of rpoA-rpoB-rpoC-rpoZ between the NdeI and HindIII sites in pT7-7 by In-Fusion Cloning Kit (Clontech). The genes were provided by

NBRP-E. coli at NIG (National Institute of Genetics, Japan). HaloTag (DHA, Promega) and His-tag fragments were attached to the 3′-end of rpoC. The Plasmid was transformed into BL21 λDE3. A single colony was inoculated into 1 litre of Luria-Bertani liquid medium (LB) containing $100\,\mu g\,ml^{-1}$ ampicillin at 37 °C until $OD_{600}$ reached 0.3–0.5. The protein production was induced by the addition of isopropyl-β-D-thiogalactoside to 0.2 mM and cells were grown overnight at 23 °C. Cells were collected by centrifugation (5 min, 5,800 g, 4 °C), rinsed with PBS and stored at $-80$ °C. To proceed with protein purification, pellets were resuspended in 80 ml lysis buffer (40 mM Tris-Cl pH 8.0, 100 mM NaCl, 10 mM EDTA and 15 mM 2-mercaptoethanol) and protease inhibitor (Roche) was added. Cells were disrupted by sonication, cleared by centrifugation (30 min, 15,300 g, 4 °C) and filtrated by the 0.22 μm syringe filter (Millipore). The protein was purified by using 5 ml HiTrap Heparin HP, 5 ml HisTrap HP and Mono Q 5/50 GL, all purchased from GE Healthcare[27]. Briefly, the cleared lysate was loaded on a 5 ml Heparin column equilibrated in buffer A (40 mM Tris-Cl pH 8.0, 1 mmM EDTA, 5% glycerol and 1 mM 2-mercaptoethanol) containing 100 mM NaCl. The column was washed with buffer A containing 300 mM NaCl and eluted with buffer A containing 600 mM NaCl. The buffer of the elutant was changed to buffer B (20 mM Tris-Cl pH 8.0 and 500 mM NaCl) containing 5 mM Imidazole by 5 ml HiTrap Desalting (GE Healthcare). The sample was loaded onto 5 ml HisTrap HP, washed with buffer B containing 20 mM Imidazole and eluted by buffer B containing 100 mM Imidazole. The buffer of the elutant was changed to buffer A containing 50 mM NaCl by 5 ml HiTrap Desalting. The sample was loaded onto Mono Q 5/50 GL and eluted by buffer A with 0.05–1 M NaCl gradient. The buffer of the eluted RNAP core enzyme was changed to storage buffer (40 mM Tris-Cl pH 8.0, 100 mM NaCl, 0.1 mM EDTA, 50% glycerol and 1 mM dithiothreitol) through a 100 kDa MWCO filter (Amicon Ultra, Millipore) and stored at $-80$ °C. The rpoD (σ[70]) gene was inserted in pT7-7 with amino terminus His tag. The Plasmid was transformed into BL21 λDE3. Cells were collected and disrupted as described above. The protein was purified by using TALON Metal Affinity Resin (Clontech) according to the product manual. The buffer of the eluted σ[70] was changed to the storage buffer through a 50 kDa MWCO filter (Amicon Ultra, Millipore) and stored at $-80$ °C.

**Single-molecule microscopy.** The fluorescent images were recorded by an Olympus IX71 inverted microscope. In the fastFISH experiment, illumination was provided by 532 and 640 nm laser light (Coherent). The lasers were combined into one fibre output by OBIS Galaxy (Coherent). The laser light was expanded to a diameter of 8 mm and focused by a 400 mm focal-length lens into the back focal plane of the objective (Olympus, ×60, numerical aperture = 1.49, oil). The fluorescent photons were collected with a electron-multiplying charge-coupled device (CCD) camera (Andor, DV887ECS-BV). The effective pixel size was 74 nm. The laser was steered by a piezo mirror (Physik Instrumente) and reflected by a dichroic mirror (FF01-577/690-25, Semrock) just below the objective lens. A dual-view apparatus (Hamamatsu Photonics) equipped with dichroic mirrors (Asahi Spectra) and emission filters (FF01-562/40-25, Semrock) was put in the Cy3 channel. In the Qdot-imaging experiment with fastFISH, illumination was provided by 488 and 532 nm laser light (Coherent). Other optics were the same as described above. To maintain the temperature in the chamber, we warmed the objective by a lens heater (TP-LH, Tokai Hit Co., Ltd, Japan).

**Single-molecule assay.** The fluorescence imaging experiments were performed inside a sample chamber assembled with a PEG-coated glass. The PEG-coated glass was prepared as follows. Coverslips were cleaned by low-pressure plasma for 5 min with a plasma system (Zepto, Diener Electronic, Germany). The coverslips were then placed into a freshly prepared 3% solution of N-2-(aminoethyl)-3-amino-propyl-trimethoxysilane (KBM-603, Shin-Etsu Chemical, Japan) in acetone for 45 min with gentle shaking. The amine-modified coverslips were then rinsed with MilliQ and dried by an air blower. A 20 μl drops of $200\,mg\,ml^{-1}$ PEG mixture solution (NHS-PEG-biotin and NHS-PEG were mixed at the ratio of 1:200 and dissolved in 0.45 M $K_2SO_4$ and 0.1 M $NaHCO_3$ pH 9.0) was squeezed between two coverslips and incubated for 30 min at 30 °C. The NHS-PEG-biotin (SUNBRIGHT BI-050TS, molecular weight = 5,000 Da) and NHS-PEG (SUNBRIGHT ME-50HS, molecular weight = 5,000 Da) were purchased from NOF Corporation, Japan[28]. PEG-coated coverslips were rinsed with MilliQ and dried by an air blower. To cap unreacted amine groups, the PEG-coated coverslips were squeezed with a 20 μl drop of Sulfo-NHS-Acetate solution (5 mg Sulfo-NHS-Acetate (Pierce) dissolved in 120 μl of 0.1 M $NaHCO_3$) for 10 min at room temperature[11]. The amine-capped coverslips were rinsed with MilliQ, dried by an air blower and stored at –80 °C under dry conditions. The sample chamber was assembled by sandwiching two pieces of double-sided tape between a non-treated coverslip and a PEG-coated coverslip.

All in vitro transcription assays were performed in transcription buffer (50 mM Tris-HCl pH 8.0, 100 mM KCl, 8 mM $MgCl_2$ and 0.1 mM EDTA) at the nucleotide concentration of $100[NTP]$ (1 mM GTP and UTP, 500 μM ATP and 250 μM CTP) to correct for the biased transcription rate for nucleotide species[12]. To tether a DNA to the biotinylated PEG glass surface, $50\,\mu g\,ml^{-1}$ avidin solution was added into a sample chamber and incubated for 3 min. After washing the chamber by transcription buffer, 1–2 nM DNA template was added and washed again by transcription buffer. In fastFISH experiment (Figs 1 and 4), single-

molecule observations were performed at 25 ± 0.5 °C or 30 ± 0.5 °C in transcription buffer containing 1 mM Trolox, 0.05% Tween, NTP and an oxygen scavenger system (0.11 mg ml$^{-1}$ glucose oxidase, 18 µg ml$^{-1}$ catalase and 2.3 mg ml$^{-1}$ glucose). The fastFISH experiment was performed in 50 nM self-quenched oligo probe, in which a hybridization rate is 0.2 s$^{-1}$ at 25 °C (Supplementary Fig. 1). In Qdot-tracking experiment with fastFISH (Fig. 2), a DNA template containing a *lac*UV5 or T7A1 promoter was incubated with a Qdot-labelled RNAP and sigma 70, to form an open complex in a test tube. Observations were performed at 25 ± 0.5 °C in transcription buffer containing 0.05% Tween, NTP and an oxygen scavenger system (0.11 mg ml$^{-1}$ glucose oxidase, 18 µg ml$^{-1}$ catalase, 2.3 mg ml$^{-1}$ glucose and and 0.5% 2-mercaptoethanol). All single-molecule observations were performed within 30 min. At least, the data within 30 min have shown no time dependency.

**AFM imaging.** AFM images were acquired on mica in air. The mica was pretreated by incubation of 10 mM MgCl$_2$ for 5 min. The mica was then rinsed with MilliQ and dried by an air blower. Before acquisition of AFM images, *in vitro* transcription was performed at the concentration of 10–20 nM DNA and 100 nM RNAP in transcription buffer for 30 min at room temperature. The reaction solution was diluted to 1–2 nM DNA in 100[NTP] deposition buffer (20 mM HEPES-KOH pH 7.8, 40 mM KCl, 10 mM MgCl$_2$ and 1 mM 2-mercaptoethanol) and added to freshly cleaved mica. After incubation for 2 min, the mica was rinsed with MilliQ and dried by an air blower. AFM imaging was performed with a NanoWizard 3 (JPK Instruments, Germany) operating in AC mode. Silicon cantilevers (OMCL-AC160TS) were purchased from Olympus. Images of 512 × 512 pixels were acquired with a scan size of <3 µm at a scan rate of two lines per second.

**Data analysis.** Images of 512 × 512 pixels were acquired by commercial software (Andor, SOLIS Software) at 10 Hz acquisition rate. In the fastFISH experiment, the positions of Cy5-labelled DNA and Qdot-labelled RNAP were decided by using a two-dimensional Gaussian distribution fitting[29] or radial symmetry-based particle localization[30]. The positions of DNA or RNAP were mapped onto Cy3 channel by the single-molecule high-resolution colocalization (SHREC) method[31–33]. In the single-molecule high-resolution colocalization (SHREC) method, before the experiment, fluorescent beads (Ultra Rainbow Fluorescent particles, 0.2 µm, Spherotech) were immobilized onto the glass surface and imaged by a electron-multiplying CCD camera through the Cy3 and Cy5 channels of the dual-view apparatus. The positions of the images on the camera were decided and a grid pattern of the paired positions was obtained by moving the bead using a piezo stage (Physik Instrumente). The 'cp2tform' command of MATLAB (MathWorks) produces a transformation structure by using the grid pattern. The 'tforminv' command of MATLAB maps the positions of Cy5 onto the Cy3 channel by using the transformation structure. To acquire the fluorescent intensity time course of oligo probe at the decided position, the averaged intensity of a 9 × 9 pixels region of interest (ROI) was calculated in each frame. For background correction, the averaged intensity of perimeter around the ROI was calculated and subtracted from the averaged intensity of the ROI[34]. The above data processing was performed with a laboratory-written programme in MATLAB. To obtain mRNA number distributions in Fig. 4a–j, the number of mRNA detections in 400 s was counted. The sample sizes in Fig. 4a–j and Supplementary Fig. 7 were evaluated for sufficient size based on the error bars in Fig. 4k,l. The judgement of signal and noise is finally based on visual inspection after defining the threshold (more than two times the s.d. of the background noise) in each time trace. We excluded traces showing unclear or multistep changes of fluorescence intensity in the analysis, because these traces do not monitor single-molecule events. In the Qdot-tracking experiment, the stage drift was corrected by tracking Qdots nonspecifically binding to the glass surface. The Qdots binding to the surface nonspecifically were distinguished based on an identity of the trace. The trace of stage drift was calculated by averaging traces of at least three nonspecifically bound Qdots. These analyses were performed with a laboratory-written programme in LabVIEW (National Instruments). As we excluded traces of nonspecifically binding Qdot from the analysis based on visual inspection, we cannot rule out the possibility that our analysis missed the traces of active Qdot-labelled RNAPs, which are arrested at the region adjacent to the promoter (<20 nm).

**Fitting by a Hill equation.** The plots in Fig. 4k were fit to the following equation,

$$m = m_{min} + (m_{max} - m_{min}) / \left[ 1 + \left( \frac{k_{half}}{k_I} \right)^n \right] \qquad (1)$$

Here, $m$ is the mean of mRNA number distribution, $m_{min}$ and $m_{max}$ are its minimum and maximum, respectively, $k_I$ is transcription initiation rate, $k_{half}$ is the rate when $m$ equals $(m_{max} - m_{min})/2$ and $n$ is the Hill coefficient.

The experimental data (red plot in Fig. 4k) were fit with $m_{min} = 0.3 ± 0.1$, $m_{max} = 3.9 ± 0.1$, $k_{half} = 0.017 ± 0.002$ and $n = 1.9 ± 0.2$. The simulated data (blue plot in Fig. 4k) were fit with $m_{min} = -0.14 ± 0.27$, $m_{max} = 4.4 ± 0.3$, $k_{half} = 0.0086 ± 0.0012$ and $n = 1.0 ± 0.2$ ( ± s.d.).

**Data availability.** The data that support the findings of this study are available from K.F. on request.

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

## Acknowledgements

We thank Ido Golding, David Priest and Jen-Chien Chang for commenting on earlier versions of the manuscript. We thank Kylius Wilkins for reading the manuscript. We thank members of the Yanagida laboratory for providing help with experiments. We thank NBRP-*E. coli* at National Institute of Genetics for providing plasmids. The work was supported by RIKEN.

## Author contributions

K.F. conceived the idea, prepared the samples, designed and carried out the experiments and the simulations, and wrote the manuscript. All authors discussed the results and the manuscript.

## Additional information

**Competing financial interests**: The authors declare no competing financial interests.

**How to cite this article**: Fujita, K. *et al.* Transcriptional bursting is intrinsically caused by interplay between RNA polymerases on DNA. *Nat. Commun.* **7,** 13788 doi: 10.1038/ncomms13788 (2016).

**Publisher's note**: 

