## [Peer Review File · Nature Communications]

Reviewers' comments:

Reviewer #1 (Remarks to the Author):

The authors studied transcriptional bursting based on in vitro single molecule assays and cellular automaton modeling, and concluded that RNAP arrest and rescue by trailing RNAP leads to transcriptional bursting on top of previous mechanisms such as supercoiling accumulation and promoter modulation. While the mathematical model is plausible, several key control experiments are missing in their single molecule assays.

First, for the key experiment in Figure 2, it is crucial to confirm the observed RNAP arrest is not due to nonspecific surface adsorption or other DNA constraints commonly seen in single molecule assays, especially when both ends of linear DNA are attached. For example, by switching to the stronger T7A1 promoter used in Fig 1 and 3, RNAP trailing and resuming elongation would be visualized directly, judging from the same time scale between Fig 1 and 2.

Second, it is worthwhile to show some other cases in Fig 2c. For example, some RNAP would elongate smoothly to the end and fall off, leading to RNA dissociation afterwards; some would stop early before the probing site, not generating any signal in Cy3 channel; some would resume elongation after temporary pause. In the current case in Fig 2c, any explanation why Cy3 signal disappear and reappear when RNAP is arrested all the time?

Moreover, according to the proposed model, RNA probing site would greatly affect the observed burstiness, i.e. 3' end of RNA show more burstiness than 5' end. Some evidence or analysis regarding this point would be helpful.

For the modeling part, the authors assumed that RNAPol arrest happens in a constant rate at each base pair, which is not the case according the Fig 2d.

Finally, it would be helpful to provide detailed information about biotin linker design in Fig 1 and 2 single molecule assays, where supercoiling accumulation may play a role otherwise.

Reviewer #2 (Remarks to the Author):

Fujita, Iwaki and Yanagida (FIY) proposed the stochastic cellular automaton model to explain the transcriptional bursting from their single cell experiment. In their approach, RNAP is arrest at DNA with some constant rate, and the arrested RNAPII can move with the interaction induced by the trailing RNAP(s).

Their probabilistic model applied especially to bacteria with introducing phenomenological kinetic processes and parameters were determined by the single cell experiment. Within this model, they claimed that they were capable to explain the sudden burst increase in the mRNA production only depend on the stochasticity as the rate of the RNAP injection into the promoter region.

I saw the work is reliable and well written. It contains novelty that possibly widely causes the interest for the readers of Nature Communications. I would like to recommend it to publish with some revisions. I hope the authors answer my concerns.

1)

In Eukaryotes, the burst-like phenomena of pre-mRNA near the CTCF/cohesion binding sites compare to the average intronic pre-mRNA is explained by the RNAP congestion near CTCF/cohesion binding sites, but in the work of FIY, RNAII is probabilistically arrest without introducing details of atomistic molecular mechanism.

It is well known that RNAPII velocity has dependency on the DNA sequence even for bacteria.

More specifically, the RNAPII cooperativity may be arisen by other than the probabilistic arrest-rescue process of FIY.

-Why can be the probabilistic arrest-rescue process of FIY dominant, compare to other factors?

-What process guarantees that the minimal component process is enough to consider for the transcription in bacteria?

-Is it fine without considering the material controlling RNAP such as the enhancer in Eukaryotes, or the gyrase in the bacteria as in Ref. 9?

-Are the parameters of the arrest-rescue process of FIY truly independent from the DNA sequence?

I hope the authors mention these points more from the character of the RNAPII flow processes as well as the experimental facts. Generally, the justification of existence of the arrest-rescue process looks need to be commented more from their experiments.

2)

Minor points: I think it is easier for the reader by explaining more on the following point.

FIY wrote: When RNAP moves from the 71th and 72th box, the nascent mRNA is "visualized" and counted. RNAP appears in the first box at a rate of k_I .

Why did authors choose the 71th and 72th?

Response to Reviewers' comments on Manuscript titled "Transcriptional bursting is intrinsically caused by interplay between RNA polymerases on DNA" by K. Fujita et al.

We thank reviewers for their careful reading of the manuscript and constructive recommendations.

Reviewer #1

First, for the key experiment in Figure 2, it is crucial to confirm the observed RNAP arrest is not due to nonspecific surface adsorption or other DNA constraints commonly seen in single molecule assays, especially when both ends of linear DNA are attached. For example, by switching to the stronger T7A1 promoter used in Fig 1 and 3, RNAP trailing and resuming elongation would be visualized directly, judging from the same time scale between Fig 1 and 2.

Reply: We concluded that the experimental geometry in Figure 2 does not impede the elongation process because the velocity of elongation is consistent with the previous report as mentioned in the manuscript and Supplementary Figure 2. Therefore, we assumed that the permanent pause in Figure 2c is not caused by interaction with the glass surface, but is due to the arrest state. Additionally, based on the reviewer's comment and in order to confirm the pushing effect by the trailing RNAPs, we performed the same experiment at 300 nM non-labeled RNAP concentration, however, without switching to the T7A1 promoter. The reason why we used the lacUV5 promoter here and in Figure 2 is because the promoter clearance from the T7A1 promoter is too fast to effectively detect the first position of RNAP before elongation under microscope (After manually adding nucleotide, RNAP often starts elongation before recording starts). Therefore, we used the same template (single-target in Fig. 2a) with 300 nM non-labeled RNAP. The Qdot labeled RNAP is binding to the promoter, and then by adding nucleotide it starts elongation, while non-labeled RNAPs can bind to the promoter after promoter clearance of the Qdot labeled RNAP. We found that the histogram of displacement varied as shown below (N=44).

Compared with Figure 2d, the peak moved to the right, suggesting that the arrested RNAP is in fact rescued by trailing non-labeled RNAPs. However, one technical difficulty of this experiment is that transcription initiation, especially open complex formation, largely depends on the flexibility of DNA because the upstream region of the promoter engulfs and stabilizes RNAP and accelerates (up to ~60 fold) transcription initiation (see for example, ref. Davis et al., “Real-time footprinting of DNA in the first kinetically significant intermediate in open complex formation by Escherichia coli RNA polymerase” *Proc Natl Acad Sci U S A* **104**, 7833-8 (2007)). Therefore, we assume that transcription initiation rate is relatively low on the stretched DNA in Figure 2b (compared to the setup in Figure 1) and it is difficult to quantitatively evaluate the transcription initiation rate and the pushing effect of a trailing RNAP in this experimental setup. Therefore, we would like to avoid directly comparing the kinetics of arrest and rescue between Figure 1 and Figure 2.

Second, it is worthwhile to show some other cases in Fig 2c. For example, some RNAP would elongate smoothly to the end and fall off, leading to RNA dissociation afterwards; some would stop early before the probing site, not generating any signal in Cy3 channel; some would resume elongation after temporary pause. In the current case in Fig 2c, any explanation why Cy3 signal disappear and reappear when RNAP is arrested all the time?

Reply: We rarely found RNAPs which dissociate from the DNA template after elongation (found in just a few tens of traces of the experiment). However, for example, the following trace shows only one oligo probe binding in the six-target DNA template because RNAP was arrested at the position between the first and second oligo probe binding site.

Also, the following trace shows a temporary pause.

However, here as well as in the manuscript, we avoid analyzing the temporary pause or movement during elongation in detail (positions or kinetics) because we think this experiment does not have sufficient spatial resolution. Nonetheless, it is clear that RNAP pause permanently (at least in the recording time) during elongation and that the arrest frequency is modulated by trailing RNAPs as suggested by the above histogram.

Regarding the reason why Cy3 signal disappears and reappears in Figure 2c, it is likely because the oligo probe binding is a Poisson process and the distribution of binding time can be described by a single exponential with a rate of 0.2 s^{-1} , therefore it is possible that some bindings take 20-30 s as shown in Figure 2c. Please note that the trace in Figure 2c was obtained with the six-target DNA template including the six binding sites for the oligo probe (To clarify this point, we revised the legend of Figure 2). We assume that a Cy3 signal disappears because of photo bleaching.

Moreover, according to the proposed model, RNA probing site would greatly affect the observed burstiness, i.e. 3' end of RNA show more burstiness than 5' end. Some evidence or analysis regarding this point would be helpful.

Reply: In the model of Figure 5, we set a rule that when RNAP moves from the 71th and 72th box, the nascent mRNA is "visualized" and counted as described in the main text. We chose the 71th and 72th box because the distance between the transcription start site and the end of the oligo probe binding site is 57 bp (Fig. 1b) and the additional elongation of 14 bp is required to "expose" the nascent mRNA (also see our reply to the second comment of the reviewer #2 and the revised last paragraph of the Results). And in fact the oligo probe binding site (the "visualization" point) in the model greatly affects the result as suggested by the reviewer. However, when it comes to experimentally testing the change, our concern about shifting the position of the oligo probe binding site downstream in the DNA template is that the mRNA transcribed before the oligo probe binding site would be longer and may compete with the oligo probe and impede the mRNA detection by it. In fact, mRNA easily forms the secondary structure as discussed in the original paper of fastFISH (ref. 11 in the revised manuscript). Therefore, if we perform the same experiment with a different position of the oligo probe binding site, it would be difficult to directly compare the results. Therefore, we would like to avoid the experiment because of the technical difficulty.

For the modeling part, the authors assumed that RNAPol arrest happens in a constant rate at each base pair, which is not the case according the Fig 2d.

Reply: As mentioned by the reviewer, we assumed that arrest rate does not depend on the sequence and position on the DNA template in this study. One of the reasons for this is that we were not able to identify clear differences in the histogram of displacement (Fig. 2d) when a different sequence was used in a preliminary experiment. If our assumption is true, theoretically the histogram of displacement in Figure 2d should be described by single exponential as shown below.

Here, we simulated the histogram of displacement with $k_A = 0.2 \text{ (s}^{-1}\text{)}$ (the value is taken from the Results), which can be described by a single exponential (blue line).

On the other hand, actually Fig 2d appears to be a single peak. However, because we excluded traces of the non-specifically binding Qdot from the analysis based on visual inspection, we cannot rule out a possibility that our analysis missed the traces of active Qdot-labelled RNAPs which are arrested at the region immediately adjacent to the promoter (We added the description about the limitations of this experiment in the legend and the Methods of the revised manuscript).

Therefore, we cannot clearly answer whether the arrest rate depends on the sequence of DNA or not, in this study. We think the question should be addressed by an *in vitro* single molecule experiment with better spatial resolution and/or high throughput techniques in the future.

Finally, it would be helpful to provide detailed information about biotin linker design in Fig 1 and 2 single molecule assays, where supercoiling accumulation may play a role otherwise.

Reply: We added structural information of the oligo modifications, including biotin linker, in the Methods according to the reviewer's comment.

Reviewer #2

1) In Eukaryotes, the burst-like phenomena of pre-mRNA near the CTCF/cohesion binding sites compare to the average intronic pre-mRNA is explained by the RNAP

congestion near CTCF/cohesion binding sites, but in the work of FIY, RNAII is probabilistically arrest without introducing details of atomistic molecular mechanism.

It is well known that RNAPII velocity has dependency on the DNA sequence even for bacteria. More specifically, the RNAPII cooperativity may be arisen by other than the probabilistic arrest-rescue process of FIY.

-Why can be the probabilistic arrest-rescue process of FIY dominant, compare to other factors?

-What process guarantees that the minimal component process is enough to consider for the transcription in bacteria?

-Is it fine without considering the material controlling RNAP such as the enhancer in Eukaryotes, or the gyrase in the bacteria as in Ref. 9?

-Are the parameters of the arrest-rescue process of FIY truly independent from the DNA sequence?

Reply: First, although we argued that transcriptional bursting is intrinsic to RNAP and DNA, transcriptional regulation is far more complicated especially in eukaryotic cells. Here, we note that transcriptional bursting is different from transcriptional regulation. While transcriptional bursting is noise, generally, transcriptional regulation should be precisely controlled in space and time like differentiation of multicellular organisms. It is known that enhancers contribute to the precise control by interacting various regulatory proteins, and some of them may also contribute to transcriptional bursting. One of the difficulties of *in vivo* experiments is that it is not easy to distinguish between transcriptional noise and regulation. In this study, in order to focus on the noise, we took a bottom-up approach to reproduce transcriptional bursting by as few components as possible. The quantitative consistency between our simulated and experimental data, and between our data and the previous *in vivo* data (Fig. 4l and Supplementary Fig. 5) suggests that the arrest-rescue process is the dominant source of transcriptional bursting in bacteria. On the other hand, we need to mention the limitations of our *in vitro* experiments (this is also associated with our reply to the reviewer #1's comment: "*For the modeling part, the authors assumed that RNAPol arrest happens in a constant rate at each base pair, which is not the case according the Fig 2d.*"). For example, although we were not able to find

clear changes of the histogram of displacement (Fig. 2d) when the sequence was changed in a preliminary experiment, we cannot rule out a possibility that our experiment does not have sufficient spatial resolution to clearly detect the sequence dependence of the arrest rate (in other words, the precise position of arrest in the context of sequence). Therefore, although we agree that the sequence dependency of velocity is an important question, it should be addressed by an *in vitro* single molecule experiment with better spatial resolution and/or high throughput techniques.

Regarding enhancers or gyrase, it is likely that such regulatory factors affect the kinetics of transcriptional bursting through control of the transcriptional initiation rate and arrest rate as we described in the Discussion of the manuscript, although we cannot quantitatively characterize their effect in this study. On the other hand, we question whether the binding and unbinding of regulatory factors such as enhancers and gyrase can fully explain the kinetics of transcriptional bursting observed *in vivo*. For example, k_{off} in the Figure 7a is a largely changeable parameter as estimated in ref. 6. In the model where the enhancers or gyrase turns on the gene, the k_{off} corresponds to their unbinding rate, which is generally constant as long as the binding sequence is unchanged. In other words, k_{off} in the models must be gene-specific, which is contrary to the global feature of transcriptional bursting observed *in vivo*. On the other hand, to explain the gene-specific feature of transcriptional bursting, for example, as observed in yeast (ref. 8 in the revised manuscript), such regulatory proteins may be required.

Additionally, the model where the RNAP congestion near CTCF/cohesion binding sites causes transcriptional bursting, seems to be fundamentally different from the model where the binding and unbinding of regulatory proteins directly corresponds to k_{on} and k_{off} . Our model may be compatible with the CTCF/cohesion model in that it focusses on the RNAP movement on DNA, although the agreement of these models should be quantitatively tested, ideally by direct observation of the flow of RNA polymerases *in vitro* or *in vivo*.

Based on the above, we added two paragraphs in the Discussion in the revised manuscript.

2) *Minor points: I think it is easier for the reader by explaining more on the following point.*

FIY wrote: When RNAP moves from the 71th and 72th box, the nascent mRNA is "visualized" and counted. RNAP appears in the first box at a rate of kl.

Why did authors choose the 71th and 72th?

Reply: We choose the 71th and 72th because we assumed that RNAP prevents the oligo probe from access to the nascent mRNA from its 3' end to -14 when the incoming nucleotide is denoted "+1" (ref. 19 in the revised manuscript). Therefore, according to the reviewer, we added a description in the last paragraph of the Results (subheading "Analysis of bursting by the cellular automaton model").

REVIEWERS' COMMENTS:

Reviewer #1 (Remarks to the Author):

The authors have addressed all the questions and concerns. Some of the answers may be presented in the main text or supplement for future readers.

Reviewer #2 (Remarks to the Author):

The responses of the authors to my questions are quite satisfactory to me.
The revised manuscript is carefully written reflecting their response, and is also fine to me.
I would like to recommend this manuscript for publication as is.